# Therapeutic Effects of Human Amniotic Epithelial Stem Cells in a Transgenic Mouse Model of Alzheimer’s Disease

**DOI:** 10.3390/ijms21072658

**Published:** 2020-04-10

**Authors:** Ka Young Kim, Yoo-Hun Suh, Keun-A Chang

**Affiliations:** 1Department of Nursing, College of Nursing, Gachon University, Incheon 21936, Korea; kykim@gachon.ac.kr; 2Neuroscience Research Institute, Gachon University, Incheon 21565, Korea; yhsuh@gachon.ac.kr; 3Department of Pharmacology, College of Medicine, Seoul National University, Seoul 03080, Korea; 4Department of Pharmacology, College of Medicine, Gachon University, Incheon 21999, Korea; 5Department of Health Sciences and Technology, Gachon Advanced Institute for Health Sciences and Technology (GAIHST), Gachon University, Incheon 21999, Korea

**Keywords:** Alzheimer’s disease, Tg2576 mice, human amniotic epithelial stem cells, amyloid plaques, learning and memory

## Abstract

Alzheimer’s disease (AD), a progressive neurodegenerative disorder, is characterized clinically by cognitive decline and pathologically by the development of amyloid plaques. AD is the most common cause of dementia among older people. However, there is currently no cure for AD. In this study, we aimed to elucidate the therapeutic effects of human amniotic epithelial stem cells (hAESCs) in a transgenic mouse model of AD. Tg2576 transgenic (Tg) mice underwent behavioral tests, namely the Morris water maze and Y-maze tests, to assess their cognitive function. In the Morris water maze test, hAESC-treated Tg mice exhibited significantly shorter escape latencies than vehicle-treated Tg mice. In the Y-maze test, hAESC-treated Tg mice exhibited significantly higher rate of spontaneous alteration than vehicle-treated Tg mice, while the total number of arm entries did not differ between the groups. Furthermore, Congo red staining revealed that hAESCs injection reduced the number of amyloid plaques present in the brains of Tg mice. Finally, beta-secretase (BACE) activity was significantly decreased in Tg mice at 60 min after hAESCs injection. In this study, we found that intracerebral injection of hAESCs alleviated cognitive impairment in a Tg2576 mouse model of AD. Our results indicate that hAESCs injection reduced amyloid plaques caused by reduced BACE activity. These results indicate that hAESCs may be a useful therapeutic agent for the treatment of AD-related memory impairment.

## 1. Introduction

Alzheimer’s disease (AD), a progressive neurodegenerative disorder, is characterized by memory loss and cognitive decline sufficient to interfere with daily life. The major pathological hallmarks of AD are senile plaques consisting of amyloid fibrils and neurofibrillary tangles made of hyperphosphorylated tau protein [1]. AD is the most common cause of dementia among older adults. According to the Centers for Disease Control and Prevention, the prevalence of AD doubles every five years after the age of 65 [2]. Despite this, there is currently no cure for AD.

However, drug therapies such as acetylcholinesterase inhibitors and N-methyl-D-aspartate antagonists are used for prevention and treatment to slow the progression of AD and reduce the symptoms of the disease [3,4]. Furthermore, novel therapeutic approaches such as stem cell-based and nano-based therapies are currently being studied and trialed for the treatment and management of AD [5,6,7]. Stem cell-based therapies have already shown promising therapeutic potential in the treatment of many serious diseases such as liver diseases, diabetes, and cardiovascular diseases [8,9,10]. In particular, recent studies have focused on stem cell-based therapies that promote the repopulation or regeneration of degenerating neuronal networks in AD [9,11,12]. Traditionally, stem cell therapy for AD has involved embryonic stem cells and mesenchymal stem cells from adult somatic tissues due to their pluripotency and immunomodulatory abilities, respectively [8,13]. Recently, human amniotic epithelial stem cells (hAESCs), a type of stem cell extracted from the innermost layer of the placenta, have been studied in humans and animals [11,14,15]. It has been reported that hAESCs have both pluripotent and immunomodulatory properties in vitro and in vivo [8,14]. Moreover, hAESCs can be readily obtained and isolated from the placenta and are non-tumorigenic upon transplantation. They are almost entirely free from ethical and legal considerations [10,14,16]. In previous studies, hAESCs have been differentiated into neurons, glial cells, cardiomyocytes, and hepatocytes [8,10,16,17,18,19,20]. In particular, recent studies have suggested that hAESCs are a promising candidate for the treatment of neurological diseases [18,21]. However, the mechanism by which hAESCs exert their therapeutic effect in AD remains unclear.

Thus, we aimed to elucidate the therapeutic effects of hAESCs in a transgenic mouse model of AD.

## 2. Results

### 2.1. Intracerebral Transplantation of hAESCs into Tg2576 Transgenic Mice

This study included four groups of mice: Tg2576 transgenic (Tg) mice and age-matched wild-type (WT) mice treated with either vehicle or hAESCs. These groups were named WT-vehicle, WT-hAESC, Tg-vehicle, and Tg-hAESC. When they were 11 months old, WT and Tg mice received bilateral intracerebral injections of vehicle or hAESCs at a single timepoint (Figure 1A). Three months later, all four groups underwent behavioral testing, including the Morris water maze and Y-maze tests. One month later, when the mice were 15 months old, pathological and molecular studies were performed. The intracerebral injection sites were in the dentate gyri of the bilateral hippocampus at the following stereotaxic coordinates relative to bregma: anteroposterior (AP) = −0.15 mm, mediolateral (ML) = ±0.13 mm, dorsoventral (DV) = −0.19 mm (Figure 1B).

### 2.2. Transplantation of hAESCs Alleviates Cognitive Deficits in Tg2576 Alzheimer’s Disease Transgenic Mice

To determine whether hAESCs transplantation alleviated cognitive deficits, we performed behavioral tests, including the Morris water maze and Y-maze tests, 3 months after intracerebral injection of hAESCs. In the Morris water maze test, which assesses spatial learning and memory, Tg mice displayed longer escape latencies than WT mice (Day, F = 32.96, *p* < 0.0001; Group, F = 12.84, *p* < 0.0001; Day x Group, F = 5.323, *p* < 0.0001). However, on day 6, escape latencies were significantly decreased in the Tg-hAESC group compared with the Tg-vehicle group (Tg-hAESC vs. Tg-vehicle, *p* = 0.0335; WT-vehicle vs. Tg-vehicle *p* < 0.0001 vs.; WT- hAESC vs. Tg-vehicle *p* < 0.0001) (Figure 2A). No significant difference was between the WT-vehicle and WT-hAESC groups A probe trial was performed 48 h after the final training trial to assess whether the mice had memorized the position of the platform (zone 4) (Zone, F = 31.10, *p* < 0.0001; Group, F = 1.138, *p* = 0.3420; Zone x Group, F = 3.337, *p* = 0.0009) (Figure 2B). The Tg-hAESC group displayed significantly recovered latency times compared with the Tg-vehicle group, while the latency times of the Tg-hAESC group were similar to those of the WT groups (Tg-hAESC vs. Tg-vehicle, *p* = 0.0340; WT-vehicle vs. Tg-vehicle, *p* = 0.0019; WT-hAESC vs. Tg-vehicle, *p* = 0.0024). The Tg-hAESC group spent significantly more time in zone 4 (PF) than in the other three zones (zones 1–3) (Figure 2C), as did the WT groups. In the Y-maze test, which assesses working memory, the Tg-hAESC group displayed significantly increased rates of spontaneous alternation compared with the Tg-vehicle group (F= 4.698, *p* = 0.0068) (Figure 2D), but the total number of arm entries did not differ between the groups (Figure 2E).

### 2.3. Transplantation of hAESCs Reduces Amyloid Burden in Tg2576 Alzheimer’s Disease Transgenic Mice

We performed Congo red staining to examine amyloid burden in the brains of hAESC- and vehicle-treated WT and Tg mice. hAESCs transplantation reduced the number of amyloid plaques in the brains of Tg2576 mice (Figure 3A). The number of amyloid plaques in the cortex (prefrontal and entorhinal cortex), hippocampus and the total number of plaques were determined. In the cortex and hippocampus, significantly fewer plaques were observed in the Tg-hAESC group than in the Tg-vehicle group (hippocampus, t = 2.100, *p* = 0.0495; cortex, t = 2.977, *p* = 0.0090). Moreover, the total number of plaques observed was significantly lower in the Tg-hAESC group than in the Tg-vehicle group (total, t = 3.344, *p* = 0.0037) (Figure 3B). Beta-secretase (BACE) activity was also assessed to understand the mechanism underlying the hAESCs transplantation-induced reduction in amyloid plaques number in Tg2576 mice. BACE activity increased over time in all four groups; however, the Tg-hAESC group showed decreased levels of BACE activity compared with the Tg-vehicle group. Treatment with hAESCs significantly decreased BACE activity in Tg2576 mice at 60 min after treatment (t = 4.222, *p* = 0.0007) (Figure 3C).

## 3. Discussion

In this study, we found that intracerebral injection of hAESCs alleviated cognitive impairment in a transgenic mouse model of AD by suppressing BACE activity and reducing amyloid burden.

When exposed to exogenous growth factors, hAESCs are able to differentiate into cells of the ectoderm, mesoderm, and endoderm. They have therapeutic potential for treating various diseases of the skin, liver, kidneys, musculoskeletal system, and central nervous system [8,14,18,19]. Furthermore, it has been reported that hAESCs can promote the regeneration of tendons, bone, and articular cartilage in large animals [14]. Moreover, hAESCs are able to differentiate into insulin-secreting pancreatic β-like cells and surfactant-producing alveolar epithelial cells [16,17,18,22,23]. It has also been reported that hAESCs can promote liver regeneration by reducing hepatic inflammation and fibrosis. These cells secrete growth factors that promote hepatocyte proliferation, exert antifibrotic actions, and induce extracellular matrix degradation, all of which have been implicated in hAESC-induced liver regeneration [20]. In the central nervous system, hAESCs promote neural cell survival and regeneration, repair damaged neurons, and reconstruct impaired neural networks [24,25]. In a previous study, hAESCs improved spatial memory in double-transgenic mice by increasing acetylcholine levels and promoting the survival of cholinergic neurites in the hippocampus [24]. In addition, human placenta amniotic membrane-derived mesenchymal stem cells have been shown to reduce amyloid deposition in the brains of C57BL/6J-APP transgenic mice via oxidative stress regulation [26].

Cognitive decline is the major clinical manifestation of AD [22]. Cognitive impairment is also a feature of neurodevelopmental disorders such as intellectual disability, autism spectrum disorder, attention deficit hyperactivity disorder, and learning disabilities in both children and older adults [23]. Amyloid-β (Aβ) deposition is one of the major pathological characteristics of AD and is closely associated with cognitive impairment [27]. The pathophysiological association between Aβ aggregation and cognition has been elucidated using functional neuroimaging [27]. Fibrillary Aβ accumulation, indicated by increased cortical uptake of radio-labeled amyloid tracers on amyloid-positron emission tomography (PET) scans, and reduced levels of Aβ in cerebrospinal fluid have been identified as diagnostic biomarkers of AD [27,28,29]. AD patients with mild cognitive impairment exhibit increased Aβ deposition on PET scans, which has been shown to be related to memory deficits [30,31,32,33,34]. In this study, behavioral tests were used to demonstrate that transplantation of hAESCs into Tg2576 mice alleviated their spatial learning and memory impairments. The Tg2576 mouse model expresses mutant human amyloid precursor protein (APP) containing the Swedish (K670N/M671L) mutation and is widely used to evaluate AD-related phenotypes and behaviors. In this model, a rapid increase in Aβ42 levels begins at 6 months of age, amyloid plaques are formed at 9–12 months of age, and memory deficits are apparent from 12 months of age [35,36].

In this study, we demonstrated that a reduction in BACE activity may be the mechanism by which hAESCs transplantation reduces the number of amyloid plaques in Tg2576 transgenic mice. BACE, also known as beta-site amyloid precursor protein cleaving enzyme, has been implicated in the formation of the Aβ42 peptide found within amyloid plaques [37]. BACE inhibition is known to prevent the production of Aβ in animals and may be a potential therapeutic approach for the treatment of patients with AD [38].

Ultimately, we showed that intracerebral transplantation of hAESCs alleviated cognitive impairment in Tg2576 AD mice. Although our results indicated that this was mediated by a reduction in amyloid burden, further studies are required to fully elucidate the mechanisms involved. We believe that hAESCs may be a useful therapeutic agent for the treatment of AD-related memory impairment.

## 4. Materials and Methods

### 4.1. Animals

Tg2576 mice expressing mutant human APP containing the Swedish (K670N/M671L) mutation were obtained from Taconic Farms (Germantown, NY, USA). Tg2576 males were mated with C57B16/SJL F1 females as reported in a previous study [36]. Only male mice were used in this study. All mice were genotyped by polymerase chain reaction analysis of tail DNA. This study included four groups of mice (10–15 mice per group). Eleven-month-old WT and Tg2576 mice were used in this study. All animal procedures were approved by the Institutional Animal Care and Use Committee of Seoul National University (IACUC No. SNU- 091208-1) and performed according to the National Institutes of Health Guidelines for the Humane Treatment of Animals.

### 4.2. Preparation of hAESCs

With approval from the Institutional Review Board of Seoul National University (IRB No. C-0809-009-255), hAESCs at passage 3 were provided by the Stem Cell Research Institute of Biostar Inc. (Seoul, Republic of Korea). Human placentas (*n* = 5) were obtained from healthy women after vaginal delivery or cesarean section. Informed consent was obtained from each donor. The amnion layer was mechanically separated from the placenta and washed several times with Hank’s balanced salt solution (without calcium or magnesium) to eliminate blood. The amniotic tissues were then digested with 10 mL trypsin–ethylenediaminetetraacetic acid (Gibco, Gaithersburg, MD, USA) under gentle agitation for 45 min at 37 °C. After the digested tissues were filtered through a 100-μm nylon sieve (Fisher Scientific, Hampton, NH, USA), they were centrifuged at 470 x *g* for 5 min. The pellet was then resuspended in amniotic epithelial cell media (Biostar Inc., Seoul, Korea) containing 10% fetal bovine serum (Gibco, Gaithersburg, MD, USA). Subsequently, the cell fractions were incubated at 37 °C and 5% CO_2_ and the medium was changed every 4–5 days. Cells at passage 3 were used for the experiments. Cell viability estimated using a trypan blue exclusion assay was above 95% prior to cell transplantation [39]. Bacterial, fungal, or mycoplasma contamination was not observed in any of the cells tested. All procedures involved in the preparation of hAESCs were performed under conditions that complied with good manufacturing practice requirements.

### 4.3. Behavioral Tests

We performed the Morris water maze test to assess spatial memory and the Y-maze to assess working memory and exploration behavior, based on previously described methods [40]. The Morris water maze test was performed 3 months after hAESCs injection. Training trials were performed 3 times a day for six consecutive days. The animal was placed in a different quadrant for each trial on a given day. An invisible platform was placed 1 cm below the surface of the water in a circular water tank. The pool was divided into four equal quadrants. Data were automatically collected using video tracking software (EthoVision, Noldus Information Technology, Wageningen, Netherlands). A single 60-s probe trial was conducted 48 h after the final training trial. The time spent in the quadrant that previously contained the platform was recorded. The Y-maze consisted of three arms: A, B, and C. The mice were placed in the maze for 8 min, and the number of times their tail entered each arm was recorded. The number of times that the animal entered all three branches consecutively without reentering a single branch was recorded as the number of alternations. The animal’s rate of spontaneous alteration (%) was calculated according to the following formula: Rate of spontaneous alternation (%) = [(number of alternations) / (total number of arm entries-2)] x 100.

### 4.4. Collection of Brain Tissue

After the behavioral tests, the mice were anesthetized with a mixture of Zoletil (12.5 mg/kg) and Rompun (17.5 mg/kg) and immediately cardiac-perfused with PBS containing heparin. One hemisphere was fixed in 4% paraformaldehyde solution for 24 hours, incubated in 30% sucrose solution for 72 hours at 4 °C and then sequential 25μm coronal sections were taken on a cryostat (Cryotome, Thermo electron cooperation, Waltham, MA, USA) and stored at 4 °C. The other hemisphere was directly incubated at −70 °C. 

### 4.5. Congo Red Staining

Hydrated sections were incubated in a freshly prepared alkaline, alcoholic, saturated sodium chloride solution (2.5 mM NaOH in 80% reagent-grade alcohol) for 20 min at room temperature (20–22 °C). The sections were then incubated in 0.5% (w/v) Congo red (Sigma, St. Louis, MO, USA) in an alkaline, alcoholic, saturated sodium chloride solution for 30 min at room temperature. Following this, the sections were washed in distilled water and counterstained with hematoxylin for 1 min. The sections were then rinsed in ascending grades of ethanol ending with 100% reagent-grade ethanol, cleared in xylene, and coverslipped using Permount mounting medium (Fisher Scientific, Miami, OK, USA). Each slide comprised ten brain sections containing the hippocampal region, and the number of plaques in the cortical (prefrontal and entorhinal cortex) and hippocampal areas of each section was counted at 200x magnification.

### 4.6. BACE Activity

BACE activity was measured in mouse hippocampal samples using a commercial BACE activity assay kit (Abcam, London, UK). Briefly, protein was extracted using ice-cold extraction buffer, incubated on ice for 30 min, and centrifuged at 10,000 x *g* for 5 min at 4 °C. The supernatant was then collected and kept on ice. The protein content of the tissue lysates was estimated using a protein determination assay. For this assay, 50 μL of tissue lysates was added to each well followed by 50 μL of 2x reaction buffer and 2 μL of BACE substrate. The reaction was incubated in the dark at 37 °C for 1 h. Each sample was run in duplicate. Fluorescence was measured at excitation and emission wavelengths of 355 nm and 510 nm, respectively.

### 4.7. Statistical Analysis

All statistical analyses were performed using GraphPad Prism version 8.4.1 software (GraphPad Software Inc., San Diego, CA, USA). All data are expressed as the mean ± standard error of the mean. With respect to data collected in the Morris water maze test, differences between the groups were analyzed by two-way analysis of variance (ANOVA) followed by the Bonferroni’s multiple comparisons test. With respect to data collected in the Y-maze test, differences between the groups were analyzed by one-way ANOVA followed by Tukey’s multiple comparisons test. Data obtained from the Congo red staining experiments and BACE activity assay were analyzed using the unpaired t-test. A *p* value <0.05 was considered statistically significant. The exact number of mice (*n*) used for each experiment is indicated in the figure legends and in the methods section.

## Figures and Tables

**Figure 1 ijms-21-02658-f001:**
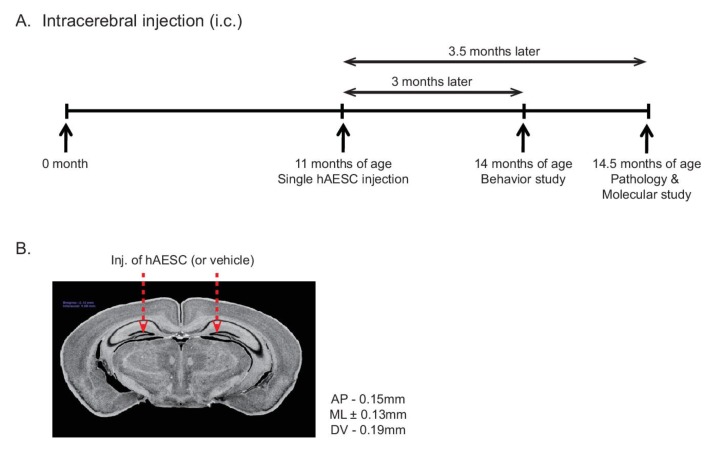
Experimental schemes. (**A**) Experimental scheme for the behavioral tests and intracerebral injections. (**B**) Stereotaxic coordinates of the intracerebral injection sites. The red arrows indicate the injected site of hAESCs (or vehicle).

**Figure 2 ijms-21-02658-f002:**
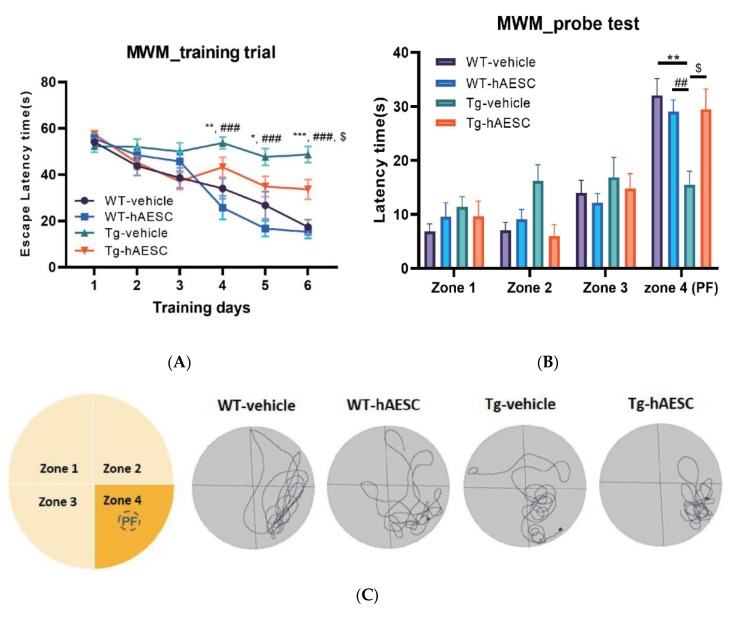
Effects of hAESCs transplantation on cognitive deficits in Tg2576 Alzheimer’s disease transgenic mice. (**A**) The Morris water maze (MWM) test was performed 3 months after intracerebral injection. Training trials were performed on 6 consecutive days, and the escape latencies of the mice were recoded. (**B**) The MWM probe test was conducted 48 h after the final training trial. (**C**) Representative swim paths in the MWM. (**D**) The total number of arm entries in the Y-maze was recorded. (**E**) The rate of spontaneous alternation in the Y-maze was calculated. All data represent the mean ± standard error of the mean (*n* = 10–15 per group). Data from the MWM test were analyzed by two-way ANOVA followed by Bonferroni’s multiple comparisons and data from the Y-maze test were analyzed by one-way ANOVA followed by Tukey’s multiple comparisons test. WT-vehicle, ^*^
*p* < 0.05, ^**^
*p* < 0.01, ^***^
*p* < 0.001; WT-hAESC, ^##^
*p* < 0.01, ^###^
*p* < 0.001; Tg-hAESC, ^$^
*p* < 0.05, ^$$^
*p* < 0.01 compared to Tg-vehicle. ANOVA, analysis of variance; hAESCs, human amniotic epithelial stem cells; Tg, transgenic.

**Figure 3 ijms-21-02658-f003:**
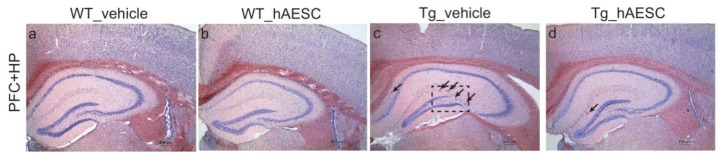
Effects of hAESCs injection on the number of amyloid plaques in the hippocampus and cortex of Tg2576 Alzheimer’s disease transgenic mice. (**A**) Hippocampal and cortical sections were stained with Congo red to detect amyloid plaques (**a**–**d**: upper panel, prefrontal cortex (PFC) and hippocampus (HP), **e**–**h**: lower panel, entorhinal cortex (EC), scale bar = 200 μm). The arrows indicate Congo red-stained amyloid plaques. Square with the dotted line contains enlarged images of the brain sections of Tg-vehicle mice (**i** & **j**) (scale bar = 100 μm). (**B**) The number of plaques in the hippocampal and cortical regions of the Tg-vehicle and Tg-hAESC groups was counted. (**C**) BACE activity levels were analyzed 60 min after injection in the Tg-vehicle and Tg-hAESC groups. All data represent the mean ± standard error of the mean (*n* = 10–15 per group). All statistical analyses were performed using the unpaired t test. ^*^
*p* < 0.05, ^**^
*p* < 0.01, ^***^
*p* < 0.001. BACE, beta-secretase; hAESC, human amniotic epithelial stem cells; Tg, transgenic.

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
