# Peer review of "Therapeutic Effects of Human Amniotic Epithelial Stem Cells in a Transgenic Mouse Model of Alzheimer’s Disease"

_ijms, 2020, doi:10.3390/ijms21072658_

Round 1

Reviewer 1 Report

In this manuscript by Kim et al., the authors found that intracerebral injection of human amniotic epithelial stem cells alleviated cognitive impairment in a Tg2576 mouse model of AD, through a reduction in amyloid plaques caused by reduced BACE activity.

The work is well presented and organized, and it can be of interest to readers and researchers which operate in the biomedical field. The figures presented by the author in the manuscript are also quite reasonable. The draft needs minor speech editing. However, the authors should improve the number of cited works, including the most recent acquisition in the specific area in introduction part. The figure 3. A is too difficult to visualize. Authors need to improve the quality of their images.

Author Response

Dear Reviewer #1

My co-authors and I would like to thank you for giving us the opportunity to submit a revised version of our manuscript. We are grateful to the editor and reviewers for their positive and constructive comments and suggestions on how to improve our manuscript entitled “Therapeutic Effects of Human Amniotic Epithelial Stem Cells in a Transgenic Mouse Model of Alzheimer’s Disease” (Manuscript ID: IJMS-757345).

We have taken into consideration the critiques of the reviewers and have revised our manuscript accordingly. Please find attached point-by-point responses to the reviewers’ comments and the revised version of our manuscript (with all changes marked in red), which we would like to submit for your consideration.

We believe that we have significantly improved the quality of our manuscript and hope it now reaches the standard of your esteemed journal. Once again, we would like to thank you for your insightful comments on our paper.

We look forward to hearing from you.

Sincerely,

Keun-A Chang, Ph.D.

Associate professor, Department of Pharmacology, College of Medicine, Gachon University

Director, Department of Basic Neuroscience, Neuroscience Research Institute, Gachon University

Reviewer #1:

  • The draft needs minor speech editing. However, the authors should improve the number of cited works, including the most recent acquisition in the specific area in introduction part.

Answer: Thank you for your comments. Although the submitted manuscript had been reviewed by a professional editing service, we appreciate that further editing was required. In response to your comment, we sought the help of an English editing company (“Editage”), and the language of the resubmitted manuscript has been checked by a native speaker. We have added 13 recent references in the introduction (page 3) and reference list (page 17) of the resubmitted manuscript.

  • The figure 3. A is too difficult to visualize. Authors need to improve the quality of their images.

Answer: Thank you for your suggestion. The images presented in Figure 3A have been improved and enlarged, and representative images of brain sections from Tg-vehicle mice have been added. Moreover, the corresponding figure legend (page 20) has been rewritten in greater detail in the resubmitted manuscript.

Reviewer 2 Report

In this paper, Kim and co-workers aime at investigating the therapeutic effects of Human Amniotic Epithelial Stem Cells (hAESC) in a Transgenic Mouse Model (Tg) of Alzheimer’s Disease (AD) .

They found out that a single injection of hAESC had positive cognitive effects on Tg mice trated with hAESC related to a placebo group of Tg AD.

These effects were replied when analyzing the levels of amyloid plaques via red COngo staining in the cortex and the activity of BACE. In both situations Tg treated with hAESC showed less levels of amyloid plaques and decreased activity of BACE suggesting that hAESC could exert his clinical effects through down-regulating BACE activity.

This is an interesting study, well designed and conciesly written.

I have some concerns that I would like to address before publication.

1) Please do report the F stastistic values of the ANOVA performed in the several analyses. 

2) Please specify better the legend of the FIgure 3 A. WHere are visible the amyloid plaques? What does correspond to the lower band of the image?Author could think to magnify the image.

3) I would like authors to discuss about the lack of effects on the hippocampal levels of amyloid plaques. Could be this related to a more advanced stage of the disease?

Author Response

Dear Reviewer #2

My co-authors and I would like to thank you for giving us the opportunity to submit a revised version of our manuscript. We are grateful to the editor and reviewers for their positive and constructive comments and suggestions on how to improve our manuscript entitled “Therapeutic Effects of Human Amniotic Epithelial Stem Cells in a Transgenic Mouse Model of Alzheimer’s Disease” (Manuscript ID: IJMS-757345).

We have taken into consideration the critiques of the reviewers and have revised our manuscript accordingly. Please find attached point-by-point responses to the reviewers’ comments and the revised version of our manuscript (with all changes marked in red), which we would like to submit for your consideration.

We believe that we have significantly improved the quality of our manuscript and hope it now reaches the standard of your esteemed journal. Once again, we would like to thank you and the reviewers for your insightful comments on our paper.

We look forward to hearing from you.

Sincerely,

Keun-A Chang, Ph.D.

Associate professor, Department of Pharmacology, College of Medicine, Gachon University

Director, Department of Basic Neuroscience, Neuroscience Research Institute, Gachon University

Reviewer #2:

1) Please do report the F statistic values of the ANOVA performed in several analyses.

Answer: Thank you for your comment. According to your suggestion, we have added F statistic values and p values into the results section (page 5-6) of the resubmitted manuscript.

2) Please specify better the legend of Figure 3 A. Where are visible the amyloid plaques? What does correspond to the lower band of the image? The author could think to magnify the image.

Answer: Thank you for your suggestion. The images presented in Figure 3A have been improved and enlarged, and representative images of brain sections from Tg-vehicle mice have been added. Moreover, the corresponding figure legend (page 20) has been rewritten in greater detail in the resubmitted manuscript.

3) I would like authors to discuss the lack of effects on the hippocampal levels of amyloid plaques. Could be this related to a more advanced stage of the disease?

Answer: Thank you very much for your valuable comment. According to your comment, we performed further statistical analysis of the amyloid plaque data using the unpaired t-test. In the original analysis, in which the data were analyzed by two-way ANOVA, the reduction in the number of plaques in the hippocampal region induced by hAESC transplantation was almost significant (the p-value obtained was extremely close to 0.05). We then reanalyzed the data according to the tissue region (cortex, hippocampus, and total) using the unpaired t-test. We found that the number of plaques in the cortex, hippocampus, and total regions was significantly lower in Tg-hAESC group than in the Tg-vehicle group (cortex, t=2.977, p=0.0090; hippocampus, t=2.100, p=0.0495; total, t=3.344, p=0.0037). We have thus added a new graph into Figure 3B and have rewritten the results section (page 6) and corresponding figure legend (page 20) in the resubmitted manuscript.

Round 2

Reviewer 2 Report

I thank the authors for their revision.

The manuscript is now ready for publication in my opinion.